# Longitudinal Examination of Which Married Young Women Use Contraception to Delay a First Pregnancy in Bihar and Uttar Pradesh, India

**DOI:** 10.3390/ijerph20156504

**Published:** 2023-08-02

**Authors:** Ilene S. Speizer, A. J. Francis Zavier, Lisa M. Calhoun, Gwyn Hainsworth, David K. Guilkey

**Affiliations:** 1Department of Maternal and Child Health, University of North Carolina at Chapel Hill, Chapel Hill, NC 27516, USA; 2Carolina Population Center, University of North Carolina at Chapel Hill, Chapel Hill, NC 27516, USA; lisa.m.calhoun@gmail.com (L.M.C.); david_guilkey@unc.edu (D.K.G.); 3Population Council, New Delhi 110003, India; fzavier@popcouncil.org; 4Bill & Melinda Gates Foundation, Seattle, WA 98109, USA; gwyn.hainsworth@gatesfoundation.org; 5Department of Economics, University of North Carolina at Chapel Hill, Chapel Hill, NC 27516, USA

**Keywords:** adolescent, young women, marriage, contraception, pregnancy, India

## Abstract

Early marriage and childbearing put young women and their babies at risk of poor health and well-being. This study uses two rounds of longitudinal data from young women ages 15–19 in 2015–2016 and followed in 2018–2019 to determine factors associated with contraceptive use before a first pregnancy among young, married women in Bihar and Uttar Pradesh, India. Discrete time hazard models were used to analyze time to first use starting from the month of marriage. Overall, use of contraception prior to a first pregnancy was low in this sample (between 12 to 20% used before a first pregnancy). Young women who reported that someone discussed the importance of delaying a first birth at the time of marriage were significantly more likely to have used a method of family planning (FP) before a first pregnancy than those who did not receive this information. Further, women who discussed FP with their husband before a first pregnancy were more likely to use contraception. Finally, among recently married young women, those who experienced pressure to have a child were less likely to use before a first pregnancy. As young women recognize the advantages of delaying a first birth and adopt FP to meet their needs, social norms around early childbearing will slowly adjust and early use to delay a first pregnancy will become more normative.

## 1. Introduction

Early marriage and childbearing, especially during the adolescent years, put young mothers and their children at risk of negative health and socioeconomic outcomes. In an analysis led by the World Health Organization [1], it was found that young mothers ages 10–19 had higher risks of eclampsia, puerperal endometritis, systemic infections, low birthweight babies, preterm delivery, and severe neonatal outcomes as compared with mothers who were ages 20–24 at the time of childbirth. Further, when women marry and begin childbearing early, they are less likely to complete schooling, engage in household economic activities, or meet their personal or professional aspirations [2,3]. 

Early marriage is high in South Asia with about 30 percent of girls marrying in childhood, that is before the age of 18 [4]. In India, at the time of the most recent National Family Health Survey (NFHS 2019–2021), 38% of women ages 20–49 married before the legal age of marriage of 18 years [5]. Notably, early marriage in India is on the decline as the percentage married by age 18 was only 23% among women ages 20–24 at the time of the 2019–2021 NFHS survey [5]. In Uttar Pradesh and Bihar, the sites of this study, 16% and 41%, respectively, of women ages 20–24 were married before 18. These differing values for early marriage correspond to differences in adolescent childbearing across the two study states with only 3% of women ages 15–19 from Uttar Pradesh but 11% from Bihar who were already mothers at the time of the 2019–2021 survey [6,7].

Early marriage is typically followed by early childbearing. In India, similar social factors influence both early marriage and early childbearing including being from rural areas, being from a less privileged caste, being from a poorer household, and a lower level of mother’s education [3]. Further, gender and social norms that govern young, married women in India and elsewhere in South Asia lead to strong social pressures for a first birth soon after marriage to prove fertility [8,9]. Married, nulliparous women are the least likely to be using contraception as shown in a multi-country study of contraception in adolescence by parity and marital status [10]. Use of contraception early in marriage to delay a first birth can help support young women to pursue their aspirations (e.g., stay in school, work, or get to know their husbands) and lead to improved health and well-being of mothers and babies. In one study from six states of India, more than half of married young women (ages 15–24) expressed a desire to postpone their first pregnancy; however, only 10% of these women had used contraception to meet this desire [8]. These young women with unmet contraceptive needs should be a priority for family planning programming to support their own, their children’s, and their families’ health and well-being [10]. Data collected as part of the “Understanding the Lives of Adolescent and Young Adults” (UDAYA) study in 2015–2016 among married adolescent women ages 15–19 in Uttar Pradesh and Bihar, India, the sites of this study, demonstrate high unmet need for family planning at 38.7% among young women in Uttar Pradesh and 45.5% among young women in Bihar [11,12].

A recent study using the 2015–2016 NFHS-4 data demonstrated that among married women in the age group 15–34, 4.7% were using a contraceptive method before a first birth; this percentage was lower than in the 2005–2006 NFHS in which 7.8% of women were using prior to a first pregnancy [13]. Study authors demonstrated that Muslim women and women of other religions were less likely to use contraception to delay a first birth while more educated women and richer women were more likely to use contraception to delay a first birth. In addition, the authors demonstrated that later age at marriage was associated with being less likely to use contraception to delay a first birth; this possibly reflects the social pressures to get pregnant soon after marriage, especially among those who marry later [13]. Finally, this study demonstrated that media exposure was associated with contraceptive use before a first birth [13]. Similarly, in an earlier study using the 2005–2006 NFHS to examine important sociodemographic determinants of pre-pregnancy contraceptive use [14], the study authors demonstrated that women who were Muslim and of other religions were less likely to use than Hindu women; higher education was associated with pre-pregnancy use; older age at marriage was associated with pre-pregnancy use; and those with weekly or daily media exposure were more likely to use before a first pregnancy [14]. These results showing key demographic differences in use before a first pregnancy may simply be reflecting common differences between users and non-users of contraception and not be specifically distinguishing factors related to pre-pregnancy use. 

In a study that collected data from young married women ages 15–24 from six Indian states, it was found that many of the young, recently married women wanted to use contraception to delay a first pregnancy (51%); however, only 10% had practiced contraception [8]. Further, in multivariate analyses among those with a demand to use contraception, the authors found a number of factors associated with use. Those who used were more educated, had greater pre-marriage awareness of family planning methods, were exposed to sexuality education, were involved in marriage-related decision-making, and felt less pressure to become pregnant soon after marriage compared to those who did not use [8]. This six-state study is rich and demonstrates the multiple associations with pre-pregnancy use among women with a reported need. Factors explored were at the individual, couple, and institutional (perceived access to healthcare) levels, and the study included additional measures of agency and exposure to family life information. The six state study results are similar to a qualitative study from two states in India (Andhra Pradesh and Telangana) that demonstrated little pre-pregnancy contraceptive use among young couples, little knowledge of contraception and sex at the time of marriage, and strong social pressures for a rapid post-marital pregnancy [9]. Understanding factors that help support young women (or couples) to use family planning (FP) to delay a first birth to meet their life aspirations as well as their fertility and FP desires can be useful for understanding how to develop programs to reduce unmet FP needs among young, recently married women and couples.

This study uses recently collected longitudinal data from young women who were married or became married in the three-year follow-up period to answer the question: What factors are associated with contraceptive use before a first pregnancy among young, married women in Bihar and Uttar Pradesh, India? A large number of factors are examined including demographic factors, age at marriage, exposure to supportive family-building messages, husband and wife communication, social pressures to have an early birth, exposure to family life education, and decision-making involvement related to the young woman’s education level.

## 2. Materials and Methods

### 2.1. Data

This study uses secondary data collected by the Population Council in India for the “Understanding the Lives of Adolescent and Young Adults” (UDAYA) study. In 2015–2016, Wave 1 data were collected among a representative sample of unmarried girls and boys (ages 10–19) and married girls ages 15–19 in the states of Bihar and Uttar Pradesh, India. A systematic, multi-stage stratified sampling design was employed to select the representative sample of participants from each state. In each state, a total of 150 primary sampling units (villages in rural areas and census wards for urban areas) were selected and following mapping and listing of households, a systematic sample of households was selected (details on sampling and segmentation can be found elsewhere [11,12]). There were three possible respondents per household: an unmarried adolescent boy (ages 10–19), an unmarried adolescent girl (ages 10–19), and a married girl (ages 15–19). In each household, only one of each type of respondent was eligible for interview for a maximum of three respondents per household. In total, the response rate was 92% for the Wave 1 survey. All participants provided informed consent to participate and for those participants who were unmarried ages 10–17, consent was also provided by a parent or guardian. All consent procedures for the primary data collection activities were reviewed and approved by the Institutional Review Board at the Population Council (protocol number 698); this secondary data analysis of de-identified data was deemed exempt from ethical review by the Institutional Review Board at the University of North Carolina (protocol number 21-2643).

Three years later (between 2018 to 2019), Wave 2 follow-up data were collected with the sample of participants who were interviewed in Wave 1 and consented to be re-interviewed. Upon data cleaning that checked the consistency of responses to age and education, the follow-up response rate for girls was 81% for Uttar Pradesh [15] and 87% for Bihar [16]. 

For this analysis, the focus is on two cohorts: Cohort A that was ages 15–19 and married (Gauna-performed) at Wave 1 and still married at Wave 2 (n = 3965); and Cohort B that was ages 15–19 and unmarried at Wave 1 and became married (Gauna-performed) by Wave 2 (n = 1497). Gauna is a North Indian practice that involves consummation of the marriage; before Gauna is performed, a woman may be married but still live at her parents’ home. In an earlier analysis with the Cohort A sample, it was found that those women who were not interviewed at Wave 2 were more likely to be from Uttar Pradesh, have no births at Wave 1, and be from urban areas [17]. The focus of this analysis is on young married women because in the India context, most childbearing and family planning use takes place within marital unions.

### 2.2. Variables

Given the multiple waves of data collection and the different statuses of the women in Cohort A and Cohort B at each wave, some variables are coded based on the Wave 1 data and others are coded based on Wave 2 data (see Appendix A Table A1, which provides information on which wave of data was used for each variable). All demographic variables in this analysis (age, education, caste, religion, wealth, and place of residence) come from the Wave 1 report. In addition, the measure of exposure to family life education comes from the Wave 1 report. The remaining independent variables related to age at marriage, decision-making about level of education, family pressure, and discussions about family planning and delaying a first pregnancy come from Wave 1 for Cohort A and from Wave 2 for Cohort B.

The key outcome variable for this analysis is timing of first use of contraception prior to a first pregnancy. For this analysis using longitudinal data, this is measured in various ways. First, all women were asked if they or their partner ever used a method to delay or prevent a pregnancy and, if yes, how many months after marriage was it when they used a method for the first time. For Cohort A women, if they had ever used contraception by Wave 1, timing of first use came from their Wave 1 report; else, we examined use and timing at Wave 2. For Cohort B women, we used Wave 2 information to inform contraceptive use and timing of first use since marriage. To determine whether first use happened before a first birth, we created a timing variable (in months) of time from marriage to first pregnancy, for those who had ever been pregnant. Since information was available on month and year of the first birth, we subtracted 9 months from that date to determine timing of first pregnancy; further, for those women who were currently pregnant with their first pregnancy (32 women at Wave 1 and 160 women at Wave 2), we subtracted 5 months to get the beginning of the pregnancy with the assumption that on average currently pregnant women were about 5 months pregnant. Using timing of first use and timing of first pregnancy, we created a dichotomous variable that determines if the month of first use is before the month of first pregnancy (coded 1 if she used before a first pregnancy and zero otherwise). Note that the 83 Cohort A women and the 12 Cohort B women who reported that they could not remember when they first used were coded as zero (did not use before first pregnancy) as this is the more conservative coding. These timing variables are used in the logistic regression models as well as the discrete time hazard models described below. 

An additional bivariate outcome that we use as a check of our timing outcome and analyses is based on a question which all women who ever used a method were asked: “Did you/your husband use (are you/your husband using) any method to delay the first pregnancy?” This variable is coded 1 if the woman reports that she (or her husband) used (or are using) a method to delay a first pregnancy and zero otherwise. For the women who reported using to delay a first pregnancy, the method used before a pregnancy was assessed; this was categorized as a hormonal/female-controlled method (pills, implant, IUD, injectable), condoms, and traditional methods (rhythm and withdrawal). 

The key independent variables for this analysis were related to husband-wife discussions, exposure to messages about delaying a first pregnancy, and pressure from family to have a child immediately. All women were asked if they ever discussed with their husband about contraception before they became pregnant the first time (yes, no, don’t know). For this variable, a response of yes is coded 1 and a response of no or don’t know is coded zero. Next, all women were asked to think back to around the time of their marriage if someone talked to them about the importance of delaying a first pregnancy (yes, no). Those women who reported yes are coded 1 and all others are coded zero. Finally, all women were asked if their in-laws or other family members put pressure on them to have a child immediately after marriage (yes, no). This variable was coded 1 if she experienced pressure and zero otherwise. 

All analyses adjusted for additional confounding variables including age at marriage, exposure to family life education, and decision-making about the woman’s education level. Age at marriage was coded as 15 or younger; 16–17; or 18 or older based on their Wave 1 (Cohort A) or Wave 2 (Cohort B) responses. Notably, none of the young women in Cohort B were married at Wave 1 and thus only a few are in the first category. Second, all women were asked at their Wave 1 interview if they ever participated in family life education (coded 1 for yes and 0 for no). Further, at each wave, women were asked who mainly decided about how much education the woman would have (the respondent only, joint with others, and others only). Information from Wave 1 is used to code Cohort A responses and information from Wave 2 is used to code Cohort B responses. Respondents were coded as having “more empowered educational decision-making” when the response was “respondent” or “joint”; these responses are coded 1 and all others are coded zero. Finally, key demographic factors that were related to contraceptive use as found in earlier analyses [8,13,14] were also included in this analysis. This included single year age at the time of the Wave 1 survey (15–19), state (Uttar Pradesh or Bihar), Wave 1 education level (none, 1–7 years, 8–9 years, 10+ years), caste (Scheduled Caste/Scheduled Tribe, Other Backward Caste, general), religion (Hindu, other), wealth quintile, and residence (urban or rural). In the full sample analyses, we included the marital status category (Cohort A or Cohort B). 

### 2.3. Analyses

All descriptive analyses employed Wave 2 weights for percentages presented; all sample sizes presented are unweighted. Analyses of use before a first pregnancy based on the bivariate categorization from the timing variables or based on reported use before a first pregnancy were undertaken using logistic regression methods and odds ratios, and 95% confidence intervals are presented for the full sample (n = 5462) and the stratified samples (Cohort A, n = 3965; Cohort B, n = 1497). These analyses were performed in Stata statistical software, version 17. 

For the outcome variable of timing of first use prior to first pregnancy, we expanded the data set such that each woman contributed months of observation from the time of first marriage until the time of first use, first pregnancy, or censuring at the time of the Wave 2 survey. With these expanded data, we used a discrete time hazard model to analyze time to first use where the starting point for the hazard was the month of marriage and then the outcome was a 0/1 variable measured at each month until first use of contraception occurred and the process is terminated (i.e., in the month in which first use occurred and we observe a “1” rather than a “0” for the outcome). A complication is that this process is terminated in the month in which a pregnancy occurs causing right censoring of the hazard for first use for those individuals who have not yet used contraception. This selection process can cause biased parameter estimates for the timing of first use hazard and the solution to this was joint estimation of the discrete time hazard for first use with a discrete time hazard for the timing of first pregnancy.

This type of joint estimation of hazard equations has been used in related studies in the past. Parametric models have been developed by Lillard [18] and there is software to implement the method [19]. We used a semiparametric method that tends to be more robust [20] to perform discrete time hazard results with correction for unobserved heterogeneity. For the discrete time hazard models, we created time dummies representing six-month intervals; these time variables indicate if contraceptive use or pregnancy experience happened in the first six months after marriage, first year after marriage, etc. Time measures were grouped into six-month intervals because some intervals between marriage and the event were quite long, and single-month dummies resulted in an extremely large number of dummies and unstable parameter estimates. The number of month groups included in the models varied slightly based on the sample and outcomes such that for the Cohort A models, we used six-month intervals with greater than three years as the reference group whereas for the Cohort B models, we only included two six-month intervals with greater than a year as the reference group. Models with Cohort B included shorter exposure time since this sample was more recently married. All hazard models were performed in Fortran. Note that all multivariate results are presented as unweighted since the hazard models which correct for heterogeneity do not use weights; in weighted analyses of the logistic results, we find similar results to what are shown here. In the multivariate analyses, significance levels of less than 0.10 are shown; however, the main findings discussed in the text are based on results that have significance levels at or below 0.05.

## 3. Results

Table 1 presents the demographic characteristics of the full sample, Cohort A, and Cohort B. At the first interview, all women were ages 15–19 but the Cohort A group were all married at Wave 1 whereas the Cohort B group were all married by Wave 2. The average age in the full sample was 17.3 years; not surprisingly, at Wave 1, Cohort A was older (mean age 18 years) than Cohort B (mean age 16.7 years). The mean age at marriage for women in Cohort A was younger (16.10) than for women in Cohort B (18.36); this makes sense given that both samples were ages 15–19 at Wave 1, but the Cohort B sample was not yet married. About two-thirds of the sample was from Bihar and one-third from Uttar Pradesh. In terms of educational level, more than half the sample had eight or more years of education and Cohort B was significantly more educated than Cohort A. No differences were observed by caste or religion between the cohorts and two-thirds were Other Backward caste and about 84% were Hindu. Wealth and place of residence also did not distinguish the groups by cohort and the overwhelming majority of the sample was from rural areas. The age at marriage categories are shown at the bottom of Table 1. In Cohort A, 85% of the sample was married before the legal age of marriage (age 18) with two-fifths married by age 15 and another two-fifths reporting marriage at age 16 or 17. In Cohort B, more than 60% were married at age 18 or older.

Table 2 presents the key independent variables for the full sample and stratified by cohort. A significantly greater percentage of respondents in Cohort B reported participating in family life education (17%) compared to Cohort A (11%); however, the overall percentage who participated in family life education is low. In addition, a significantly higher percentage of Cohort B participants reported that they had a decision-making role in determining their education level as compared to Cohort A. About a quarter of women reported that they ever discussed family planning with their husband before the first pregnancy. No difference was observed by cohort. In terms of someone talking to the women around the time of marriage about the importance of delaying a first birth, about 13% of the full sample reported that someone talked to them. A significantly greater percentage (15.4%) of women in Cohort A reported that someone talked to them compared to Cohort B (10.9%). Finally, nearly a fifth of women reported that they experienced pressure from their in-laws or another family member to have a child immediately after marriage; no difference was observed by cohort. 

For the dependent variable of whether the woman used a method of contraception before the first pregnancy (based on the timing questions), we found that 15.5% of the full sample reported using before a first pregnancy (see Table 2). This percentage was significantly higher among Cohort A women (19.8%) than among Cohort B women (11.9%). Similarly, based on the specific question on whether the woman (or her partner) used a method to delay or avoid a first pregnancy, 10.7% of the full sample reported using before a first pregnancy with a significantly higher percentage reporting the same in Cohort A (14.3%) than in Cohort B (7.7%). In both cohorts, among those women who reported use before a first pregnancy, traditional methods were the most common methods reported (greater than 50% of users) followed by condoms (about 42% of users) and hormonal/female-controlled methods (4–8%).

Table 3 presents the logistic regression odds ratios and 95% confidence intervals (CI) for the analysis of use of any contraception before a first pregnancy based on the timing questions. The key variables of interest are at the bottom of the table. In the full sample and in Cohort A and Cohort B, we found that if someone talked to the woman about delaying a first birth at the time of marriage, the respondent had significantly higher odds (between 1.5–2.1) of using a method prior to a first pregnancy than if someone did not talk to her (or she didn’t remember). Further, in the full and stratified samples, if the couple discussed FP before a first pregnancy, she had higher odds of using before a first pregnancy than among couples who did not discuss FP. Only in the Cohort B sample (i.e., recently married), we found that if she experienced pressure to have a birth from a family member, she had lower odds of using a method before a first pregnancy (OR: 0.53; 95% CI: 0.32–0.87, *p* ≤ 0.05). In addition, in the Cohort B sample, those who married at an earlier age had lower odds of using a method before a first pregnancy than those who married at an older age. Further, in this same sample, those who were older at the time of the Wave 1 survey were less likely to have used before a first pregnancy than those who were younger at Wave 1. Across the full and stratified samples, women from Bihar had lower odds of using before a first pregnancy than women from Uttar Pradesh. In Cohort A, more educated women and women in the highest wealth group had higher odds of using before a first pregnancy than less educated women and women in the middle wealth group. That said, in the Cohort B sample, women in the lowest wealth group had lower odds of pre-pregnancy use than women in the middle wealth group. Finally, in the full sample, women in Cohort B had lower odds of using before a first pregnancy than women in Cohort A.

In the analysis of reported use of contraception to delay or avoid a first pregnancy (Table 4), the results were similar to the above findings, with a few notable exceptions. In all three samples, husband-wife discussion of FP remained significant with a similar effect. In the full sample and in Cohort A, someone talking to the woman before a first pregnancy was significantly associated with the woman’s reported use to avoid or delay a first pregnancy; this was not significant in the Cohort B sample. Further, pressure to have a child remained significant in the Cohort B sample such that those who experienced pressure had lower odds of using. In the analysis of reported use before a first pregnancy, in the full sample and in Cohort A, young women who received family life education had higher odds of pre-pregnancy use than those who did not receive family life education. In the full sample and Cohort A sample, those who married earlier had lower odds to self-report using before a first pregnancy than those who married at age 18 or older. That said, in Cohort B, those who married at ages 16 or 17 had higher odds of using before a first pregnancy than those who married at age 18 or older. In the full sample and Cohort A sample, age was significant such that older women at the time of the Wave 1 survey had lower odds of using a method to delay or avoid pregnancy than younger women. Finally, as for the previous outcome, those in Cohort B had lower odds of using before a first pregnancy than those in Cohort A.

Table 5 presents the corrected discrete time hazard ratios and significance for time to first use prior to a first pregnancy based on timing since marriage, correcting for unobserved heterogeneity through joint estimation with the timing of first pregnancy equation (see Table 6 for pregnancy results). In terms of the key independent variables, the results were similar to those presented earlier. In particular, in the corrected results, those women whom someone talked to about delaying a first pregnancy had a higher probability of using a method before a first pregnancy over time than those women whom no one talked to; this was true in the full sample and both cohort samples. Further, those women who ever discussed family planning with their husband before a first pregnancy were also more likely to use before a first pregnancy in the full and cohort samples. In Cohort B only, as above, those women who experienced pressure to have a child immediately after marriage were less likely to use before a first pregnancy. The results for age at marriage were also similar to what was presented earlier. For both cohorts, those who married earlier were significantly less likely to use before a first pregnancy. Also shown for these models is time. In the full sample and cohort stratified models, women were more likely to use before a first pregnancy in the earlier periods post marriage (i.e., months 1–6, 7–12, or 13–18) than later (after 2 years). As earlier noted, women from Bihar were less likely to use while more educated women were more likely to use in each month after marriage. In the full sample, women in Cohort B were less likely to use at each month after marriage than women in Cohort A. 

Table 6 presents the discrete time hazard ratios and significance for timing of first pregnancy since marriage correcting for unobserved heterogeneity; these are the selection equation results. Over time, those women who experienced pressure to have a child were less likely to have had a child in each month conditional on not having a child prior to that month. Those women who married earlier in Cohort A were less likely to have experienced a pregnancy in each subsequent month since marriage than those who married later. In Cohort B that had a later overall age at marriage, the results were the opposite such that the ones who married earlier were more likely to have become pregnant in each month. Notably, the time since marriage variables have opposite effects between Cohorts A and Cohort B, possibly reflecting the earlier age at marriage in Cohort A. Respondents from Bihar were more likely to experience a pregnancy in each month after marriage in the full sample and in each of the cohorts. Hindu women were less likely to get pregnant in each month than were Muslim women in the full sample and Cohort A. Further in the full sample and Cohort A, urban women were more likely to get pregnant in each month than rural women. Finally, the Cohort B sample was less likely to have experienced a pregnancy in each month following marriage than the Cohort A sample. 

## 4. Discussion

This study contributes to our understanding of which married young women in two states in India use contraception prior to a first pregnancy. It is important to identify the young women who were able to adopt a method and delay childbearing as prior research has demonstrated that there are high unmet needs for contraception among young, married women [8]. In contexts like India where early marriage is common, supporting young women (and couples) to use contraception to delay a first birth can help to improve health and well-being outcomes of women, children, and families. 

Overall, use of contraception prior to a first pregnancy was the exception rather than the rule in this young, married sample from Bihar and Uttar Pradesh. In particular, we found that about 20% of women reported that the timing of their first contraceptive use was before their first pregnancy and separately 15% of women reported that they specifically used a method of contraception to delay or avoid a first pregnancy. The use levels were higher in the Cohort A sample, that is, among those women who married earlier and had been married longer at the time of data collection. Among the women who reported using a method to delay or avoid childbearing, the most common method reported (53%) was traditional methods followed by condoms (42%) and only a small percentage (5%) reported hormonal or female-controlled methods. It is interesting that even among the small number of women using a method prior to a first pregnancy, most of the methods used are traditional or less effective (condom) methods. This may be reflective of social norms that encourage immediate childbearing after marriage, which might make access to and choice of more effective methods less possible. It may also reflect a more limited selection of methods in the India context where injectables and implants have only recently been introduced into the public health system following the launch of the FP2020 program (https://fp2030.org/sites/default/files/India_FP2030_Vision_Document.pdf, accessed on 8 September 2022).

In terms of factors associated with use prior to a first pregnancy, results presented here demonstrate the importance of interpersonal relations in support of contraceptive decision-making and use. Young women who reported that someone talked to them about the importance of delaying a first birth at the time of marriage were significantly more likely to have used a method of FP before a first pregnancy than young women who did not receive this information. Further, those young women who reported that they discussed FP with their husband before they became pregnant for the first time were also more likely to use before a first pregnancy. An additional interpersonal factor that was particularly important in the Cohort B sample was experiencing pressure from a family member (mother-in-law or another family member) to have a child; this was associated with non-use of FP prior to a first pregnancy. Finally, in the Cohort A sample, those who had family life education were more likely to report that they used a method to delay or avoid a pregnancy than those who did not have family life education. The remaining important factor that was related to interpersonal relations was age at marriage. In this sample in which more than 80% of the Cohort A sample was married before 18, the legal age of marriage (and 40% in Cohort B), this was an important factor that may be related to many sexual and reproductive health outcomes. In our analysis, we found that young women who married earlier in Cohort A were less likely to use a method prior to a first pregnancy; this may reflect less decision-making autonomy among the women who were child brides. 

Our results are similar to those of earlier studies on contraceptive use to delay a first pregnancy in India and elsewhere. In the Jejeebhoy et al.’s six state study [8], the authors found that among the small percentage of young, married women using a method to avoid a first pregnancy, condoms were the most commonly used method followed by traditional methods and oral contraceptive pills in the northern and eastern regions. In the multivariate analyses, the six-state study found that among those young women who had a demand for contraception, that is they wanted to delay a first pregnancy, those who used were more educated, had greater pre-marriage awareness of family planning methods, were exposed to sexuality education, were involved in marriage-related decision-making, and felt less pressure to become pregnant soon after marriage compared with those who did not use a method [8]. Our findings were similar to these in terms of exposure to information about the benefits of family planning and pressure to have a child. Further, a recent study that used the NFHS-4 data from 2015 to 2016 demonstrated similar factors associated with the use of contraception before a first birth including religion, caste, education, wealth, age at marriage, and media exposure [13]. These results are consistent with an NFHS study from 10 years earlier that showed similar factors associated with pre-pregnancy use among women of all ages [14]. The factors found to be associated with use in these quantitative studies may simply reflect distinctions in use patterns in the study sites and not be specific to demographic factors associated with use to delay a first pregnancy. 

A recent qualitative study of young women and couples from two states in India demonstrated important gaps in young women’s (and couples’) access to information and services that are likely related to their involvement in early marriages and early childbearing [9]. In particular, study authors point out that young women and couples lack information on contraceptive options, experience significant family pressures to bear children immediately after marriage, and do not necessarily have access to a full range of contraceptive methods. These results are consistent with our study findings that show low exposure to family life education in the study sample; about a fifth of young women report experiencing pressure from a family member to have a birth; and low use of a pre-pregnancy method with most use being the traditional method use. Our analysis has some strengths and extends the earlier quantitative studies by including all young women in the analysis of which young women were using a method to delay a first birth rather than simply focusing on those who reported a desire to delay a birth, as done in the six-state study [8]. Further by using longitudinal data, we were able to examine changes in use (and pregnancy) status over the follow-up period. This also permitted including a sample who married earlier and had been married for a longer period of time (Cohort A), as well as a sample who had recently married at a later age. By using hazard models that controlled for selection bias, we were able to better assess the interpersonal factors that were associated with young women’s use of a method to delay a first pregnancy, controlling for key demographic factors associated with use. 

This study also has limitations. First, this study used retrospective reports of timing of first use and first pregnancy relative to timing of marriage, and thus a small number of women did not remember when they first used or may have misreported first use if it was not considered socially desirable to use before a first pregnancy. Further, we found distinctions between women’s reported use of contraception to delay a first pregnancy and the percentage who were estimated to use based on the timing of the events; while both are low, reported use is lower than the estimates based on the timing measures. Again, this may reflect social desirability bias around contraceptive use behaviors among young married women. Another limitation is that we did not include the full range of factors which may be related to pre-pregnancy contraceptive use including agency, empowerment, and experience of violence. While some of this information was available in the data, models that included these types of variables found no significant effects nor improvement in model fit, and thus these variables were dropped from the final models. An additional limitation with these variables was that they were not time-dependent, and thus for Cohort A, it was not possible to assess if agency or empowerment were a consequence of marriage, pregnancy or contraceptive use or influenced these behaviors. Future longitudinal studies should consider strategies to assess how these important factors are related to contraceptive use and pregnancy timing by asking questions specifically designed with this in mind. Finally, in the multivariate results of pregnancy experience, we found that those women who experienced pressure to have a child were less likely to experience a pregnancy at each month of follow-up than those who did not experience pressure. This may reflect that those who have not yet had a child were experiencing (or remembering) more pressure than those who had an early (or timely) pregnancy. With the data available, it is not possible to explain in detail this unexpected result. 

## 5. Conclusions

While broader norm change around timing of first pregnancies may take time to observe, understanding which young, married women were able to use contraception early in their marriage is important for reducing unmet need for family planning and ensuring that all young women (and couples) are able to use family planning to meet their aspirations, fertility, and FP needs. Those young women who heard about the importance of delaying a first birth around the time of marriage were the most likely to be early FP adopters, that is before a first pregnancy. This highlights the need for FP programs to include messaging, and social and behavior change interventions that promote the advantages of delayed first birth and couple communication among young married (or soon to be married) women, their partners, and gatekeepers. Intervention channels could include frontline workers, school-based programs, mass media, digital platforms, and community-based engagement activities. As more young women recognize the advantages of delaying a first birth and begin to use family planning to meet their needs, it is expected that social norms around early childbearing will slowly adjust and early use to delay a first pregnancy will become more normative.

## Figures and Tables

**Table 1 ijerph-20-06504-t001:** Descriptive characteristics of young women married at Wave 1 (Cohort A) or who became married between Wave 1 and Wave 2 (Cohort B), Bihar and Uttar Pradesh, India.

	Total (n = 5462)		Cohort A (n = 3965)		Cohort B (n = 1497)		*p*-Value
Characteristic	%	n	%	n	%	n	
Wave 1 Age (mean)	17.32 years		18.02 years		16.73 years		***
State							
Uttar Pradesh	35.66	1907	36.25	1303	35.17	604	
Bihar	64.34	3555	63.75	2662	64.83	893	
Wave 1 Education							
None	17.83	1235	26.06	1088	10.93	147	***
1–7 years education	22.59	1220	23.48	897	49.36	323	
8–9 years education	29.68	1472	25.75	997	32.97	475	
10+ years education	29.91	1535	24.70	983	34.28	552	
Caste							
Scheduled Caste/Scheduled Tribe	27.19	1514	28.49	1149	26.09	365	
Other Backward Caste	62.11	3380	61.15	2439	62.92	941	
General	10.70	568	10.37	377	10.99	191	
Religion							
Other religion	16.01	915	15.06	608	16.81	307	
Hindu	83.99	4547	84.94	3357	83.19	1190	
Wave 1 Wealth							
Lowest	17.44	888	17.05	656	17.77	232	
Low	22.43	1077	22.04	784	22.76	293	
Medium	24.65	1277	24.09	932	25.12	345	
High	22.61	1386	23.38	997	21.97	389	
Highest	12.86	834	13.44	596	12.38	238	
Residence							
Rural	90.66	3542	89.94	2527	91.26	1015	
Urban	9.34	1920	10.06	1438	8.74	482	
Age at marriage							
15 or younger	21.32	1755	41.26	1699	4.61	56	***
16–17	39.01	2187	43.47	1693	35.27	494	
18+	39.66	1520	15.27	573	60.12	947	
Marital Status Category							
Cohort A	45.61	3965					
Cohort B	54.39	1497					

Notes: All means and percentages use Wave 2 weights that adjust for non-response between Waves; sample sizes are unweighted. *** *p* ≤ 0.001 significant difference between Cohort groups.

**Table 2 ijerph-20-06504-t002:** Key independent variables among young women married at Wave 1 (Cohort A) or who became married between Wave 1 and Wave 2 (Cohort B), Bihar and Uttar Pradesh, India.

	Total (n = 5462)		Cohort A (n = 3965)		Cohort B (n = 1497)		
Characteristic	%	n	%	n	%	n	*p*-Value
Participated in family life education							
No	85.9	4763	89.09	3534	83.23	1299	***
Yes	14.1	699	10.91	431	16.77	268	
Decision making about education level							
Someone else decides	47.48	2841	55.50	2243	40.76	598	***
Respondent/joint decision	52.52	2621	44.50	1722	59.24	899	
Discussed family planning with husband before became pregnant the first time							
Never	76.60	4163	77.15	3044	76.13	1119	
Ever	23.40	1299	22.85	921	23.87	378	
Someone talked to you about the importance of delaying first pregnancy at the time of your marriage							
No	87.05	4628	84.63	3303	89.09	1325	**
Yes	12.95	834	15.37	662	10.91	172	
Experienced pressure from in-laws or other family members to have a child immediately after marriage							
No	80.11	4427	79.06	3197	81.00	1230	
Yes	19.89	1035	20.94	768	19.00	267	
Used a method of contraception before a first birth/pregnancy (based on timing questions)							
No	84.50	4482	80.24	3189	88.07	1293	***
Yes	15.50	980	19.76	776	11.93	204	
Used to delay or avoid first pregnancy (based on specific question)							
No	89.27	4784	85.69	3406	92.27	1378	***
Yes	10.73	678	14.31	559	7.73	119	
Method used before pregnancy (among those who reported use to delay/avoid first pregnancy)							
Hormonal/female-controlled	5.28	33	3.63	26	7.85	7	
Condom	42.08	287	43.17	235	40.38	52	
Traditional method	52.64	358	53.2	298	51.77	60	

Notes: All percentages use Wave 2 weights that adjust for non-response between Waves; sample sizes are unweighted; ** *p* ≤ 0.01; *** *p* ≤ 0.001 significant difference between cohort groups.

**Table 3 ijerph-20-06504-t003:** Logistic regression odds ratios (95% CI) for use of contraception before first pregnancy based on timing questions by analysis sample.

Variable	Full Sample (n = 5462)		Cohort A (n = 3965)		Cohort B (n = 1497)	
	Odds Ratio	95% CI	Odds Ratio	95% CI	Odds Ratio	95% CI
Bihar (vs. UP)	0.472	(0.40–0.55) ***	0.479	(0.40–0.57) ***	0.406	(0.28–0.59) ***
Other backward caste (vs. SC/ST)	1.020	(0.85–1.22)	1.097	(0.90–1.34)	0.743	(0.49–1.13)
General (vs. SC/ST)	0.856	(0.64–1.14)	0.914	(0.66–1.27)	0.673	(0.36–1.25)
Age (continuous)	0.931	(0.86–1.01) +	0.940	(0.86–1.03)	0.811	(0.68–0.97) *
Education 1–7 years (vs. none)	1.116	(0.85–1.46)	1.103	(0.83–1.47)	1.477	(0.60–3.64)
Education 8–9 years (vs. none)	1.210	(0.93–1.58)	1.264	(0.95–1.67)	1.214	(0.50–2.92)
Education 10+ years (vs. none)	1.681	(1.27–2.22) ***	1.733	(1.28–2.34) ***	1.872	(0.78–4.52)
Hindu (vs. Other religion)	1.245	(0.99–1.56) +	1.152	(0.89–1.49)	1.613	(0.98–2.65) +
Lowest wealth group (vs. Medium)	0.739	(0.56–0.97) *	0.831	(0.61–1.12)	0.466	(0.24–0.91) *
Low wealth group (vs. Medium)	0.873	(0.69–1.11)	0.858	(0.65–1.13)	0.968	(0.57–1.64)
High wealth group (vs. Medium)	1.024	0.83–1.27)	1.121	(0.88–1.42)	0.775	(0.47–1.27)
Highest wealth group (vs. Medium)	1.335	(1.05–1.70) *	1.376	(1.05–1.81) *	1.384	(0.81–2.37)
Urban (vs. Rural)	1.035	(0.88–1.22)	0.979	(0.81–1.18)	1.246	(0.84–1.84)
Discussed family planning with husband	3.809	(3.25–4.47) ***	2.923	(2.43–3.51) ***	10.178	(7.13–14.52) ***
Someone talked about delaying 1st birth	1.516	(1.26–1.83) ***	1.487	(1.21–1.83) ***	2.120	(1.35–3.32) ***
Experienced pressure to have child	0.851	(0.70–1.04)	0.923	(0.74–1.15)	0.531	(0.32–0.87) *
Respondent involved in education decision (vs. not)	1.112	(0.94–1.32)	1.082	(0.89–1.31)	1.320	(0.90–1.93)
Participated in family life education (vs. did not)	1.057	(0.85–1.31)	1.136	(0.88–1.46)	0.805	(0.51–1.26)
Age at marriage ≤15 (vs. 18+)	0.926	(0.72–1.19)	1.025	(0.78–1.35)	0.329	(0.10–1.11) +
Age at marriage 16–17 (vs. 18+)	0.892	(0.72–1.10)	1.009	(0.79–1.29)	0.526	(0.32–0.86) *
Cohort B vs. Cohort A (newly married vs. previously married)	0.486	(0.38–0.62) ***	N/A		N/A	

+ *p* ≤ 0.10; * *p* ≤ 0.05; *** *p* ≤ 0.001; N/A—not applicable.

**Table 4 ijerph-20-06504-t004:** Logistic regression odds ratios (95% CI) for reported use of contraception to delay or avoid a first pregnancy by analysis sample.

Variable	Full Sample (n = 5462)		Cohort A (n = 3965)		Cohort B (n = 1497)	
	Odds Ratio	95% CI	Odds Ratio	95% CI	Odds Ratio	95% CI
Bihar (vs. UP)	0.517	(0.43–0.62) ***	0.498	(0.40–0.62) ***	0.571	(0.37–0.88) *
Other backward caste (vs. SC/ST)	1.062	(0.86–1.31)	1.054	(0.83–1.33)	1.090	(0.65–1.84)
General (vs. SC/ST)	0.861	(0.62–1.20)	0.790	(0.54–1.16)	1.078	(0.52–2.25)
Age (continuous)	0.777	0.71–0.85) ***	0.713	(0.64–0.79) ***	1.209	(0.96–1.52)+
Education 1–7 years (vs. none)	1.359	(0.98–1.88) +	1.325	(0.94–1.87)	1.910	(0.64–5.67)
Education 8–9 years (vs. none)	1.141	(0.83–1.57)	1.050	(0.74–1.48)	1.839	(0.64–5.29)
Education 10+ years (vs. none)	1.622	(1.16–2.26) **	1.627	(1.13–2.33) **	1.674	(0.58–4.86)
Hindu (vs. Other religion)	1.207	(0.92–1.58)	1.218	(0.90–1.66)	1.222	(0.70–2.14)
Lowest wealth group (vs. Medium)	0.816	(0.59–1.13)	0.841	(0.58–1.210	0.678	(0.30–1.53)
Low wealth group (vs. Medium)	0.872	(0.65–1.16)	0.791	(0.57–1.10)	1.286	(0.68–2.43)
High wealth group (vs. Medium)	1.174	(0.91–1.51)	1.161	(0.88–1.53)	1.307	(0.73–2.34)
Highest wealth group (vs. Medium)	1.425	(1.08–1.89) *	1.484	(1.09–2.03) *	1.384	(0.71–2.71)
Urban (vs. Rural)	0.932	(0.77–1.13)	0.923	(0.74–1.15)	0.929	(0.58–1.48)
Discussed family planning with husband	4.991	(4.15–6.00) ***	4.358	(3.54–5.37) ***	8.867	(5.76–13.65) ***
Someone talked about delaying 1st birth	1.440	(1.16–1.78) ***	1.550	(1.23–1.96) ***	1.109	(0.63–1.95)
Experienced pressure to have child	0.668	(0.52–0.85) ***	1.484	(0.61–1.03) +	0.258	(0.12–0.55) ***
Respondent involved in education decision (vs. not)	1.091	(0.89–1.33)	0.792	(0.87–1.37)	1.029	(0.66–1.60)
Participated in family life education (vs. did not)	1.366	(1.07–1.74) *	1.090	(1.13–1.96) **	1.021	(0.61–1.72)
Age at marriage ≤15 (vs 18+)	0.572	(0.42–0.77) ***	0.485	(0.35–0.67) ***	1.371	(0.34–5.59)
Age at marriage 16–17 (vs. 18+)	0.806	(0.63–1.03) +	0.698	(0.53–0.92) **	1.948	(1.06–3.56) *
Cohort B vs. Cohort A (newly married vs. previously married)	0.265	(0.20–0.36) ***	N/A		N/A	

+ *p* ≤ 0.10; * *p* ≤ 0.05; ** *p* ≤ 0.01; *** *p* ≤ 0.001; N/A—not applicable.

**Table 5 ijerph-20-06504-t005:** Discrete time hazard ratios with correction for unobserved herterogeneity for timing of first use since marriage.

Variable	Full Sample			Cohort A		Cohort B
	Hazard Ratio	Significance	Hazard Ratio	Significance	Hazard Ratio	Significance
Bihar (vs. UP)	0.269	***	0.347	***	0.069	***
Other backward caste (vs. SC/ST)	1.133	NS	1.423	NS	0.428	*
General (vs. SC/ST)	0.598	+	0.791	NS	0.327	NS
Age (continuous)	0.720	***	0.760	**	0.387	**
Education 1–7 years (vs. none)	1.524	NS	1.779	+	5.477	*
Education 8–9 years (vs. none)	1.662	+	1.723	+	4.703	+
Education 10+ years (vs. none)	3.315	***	3.423	***	4.997	*
Hindu (vs. Other religion)	1.457	+	1.026	NS	1.223	NS
Lowest wealth group (vs. Medium)	0.527	NS	0.802	NS	0.020	*
Low wealth group (vs. Medium)	0.529	NS	0.593	+	0.399	NS
High wealth group (vs. Medium)	0.907	NS	1.192	NS	0.167	NS
Highest wealth group (vs. Medium)	1.710	NS	1.453	NS	0.876	NS
Urban (vs. Rural)	1.128	NS	1.184	NS	0.806	NS
Discussed family planning with husband	15.192	***	7.012	***	1801.077	***
Someone talked about delaying 1st birth	2.483	***	1.810	*	6.871	***
Experienced pressure to have child	0.767	NS	0.733	NS	0.062	***
Time: Months 1–6 (vs. >2 years)	14.216	***	17.800	***	6.031	**
Time: Months 7–12 (vs. >2 years)	2.309	*	2.435	*	1.966	NS
Time: Months 13–18 (vs. >2 years)	3.496	***	4.137	***	ref.	
Time: Months 19–24 (vs. >2 years)	0.695	NS	0.864	NS	ref.	
Age at marriage ≤ 15 (vs 18+)	0.432	***	0.470	**	0.005	***
Age at marriage 16–17 (vs. 18+)	0.608	*	0.677	+	0.169	**
Respondent involved in education decision (vs. not)	1.160	NS	1.059	NS	1.793	NS
Participated in family life education (vs. did not)	1.121	NS	1.251	NS	0.977	NS
Cohort B vs. Cohort A (newly married vs. previously married)	0.253	***	N/A		N/A	

NS: Not significant; + *p* ≤ 0.10; * *p* ≤ 0.05; ** *p* ≤ 0.01; *** *p* ≤ 0.001; N/A—not applicable; ref. is the reference group.

**Table 6 ijerph-20-06504-t006:** Discrete time hazard ratios with correction for unobserved heterogeneity for timing of first pregnancy since marriage.

Variable	Full Sample		Cohort A		Cohort B	
	Hazard Ratio	Significance	Hazard Ratio	Significance	Hazard Ratio	Significance
Bihar (vs. UP)	1.263	***	1.234	***	1.476	***
Other backward caste (vs. SC/ST)	0.985	NS	0.965	NS	0.966	NS
General (vs. SC/ST)	0.954	NS	0.927	NS	1.010	NS
Age (continuous)	0.951	*	0.919	***	1.184	***
Education 1–7 years (vs. none)	1.047	NS	0.996	NS	1.035	NS
Education 8–9 years (vs. none)	1.063	NS	1.114	NS	0.915	NS
Education 10+ years (vs. none)	1.031	NS	1.054	NS	0.984	NS
Hindu (vs. Other religion)	0.796	**	0.767	**	0.998	NS
Lowest wealth group (vs. Medium)	0.825	**	0.844	+	0.848	NS
Low wealth group (vs. Medium)	0.934	NS	0.954	NS	0.928	NS
High wealth group (vs. Medium)	0.987	NS	1.004	NS	0.928	NS
Highest wealth group (vs. Medium)	0.990	NS	1.039	NS	0.850	NS
Urban (vs. Rural)	1.210	***	1.217	**	1.115	NS
Discussed family planning with husband	1.011	NS	1.058	NS	0.826	*
Someone talked about delaying 1st birth	1.000	NS	1.015	NS	0.992	NS
Experienced pressure to have child	0.829	***	0.897	*	0.680	***
Time: Months 1–6 (vs. >3 years)	0.467	***	0.502	*	1.310	*
Time: Months 7–12 (vs. >3 years)	0.646	**	0.709	NS	1.519	***
Time: Months 13–18 (vs. >3 years)	0.569	***	0.615	NS	ref.	
Time: Months 19–24 (vs. >3 years)	0.817	NS	0.905	NS	ref.	
Time: Months 25–30 (vs. >3 years)	0.761	*	0.813	NS	ref.	
Time: Months 31–36 (vs. >3 years)	0.975	NS	1.020	NS	ref.	
Age at marriage ≤ 15 (vs 18+)	0.466	***	0.363	***	1.638	**
Age at marriage 16–17 (vs. 18+)	0.813	**	0.637	***	1.484	***
Respondent involved in education decision (vs. not)	0.970	NS	0.990	NS	0.939	NS
Participated in family life education (vs. did not)	0.990	NS	0.998	NS	0.921	NS
Cohort B vs. Cohort A (newly married vs. previously married)	0.778	**	N/A		N/A	

NS: Not significant; + *p* ≤ 0.10; * *p* ≤ 0.05; ** *p* ≤ 0.01; *** *p* ≤ 0.001; N/A—not applicable; ref. is the reference group.

## Data Availability

All data used for this paper are publicly available on Harvard Dataverse at: https://doi.org/10.7910/DVN/RRXQNT, accessed on 4 May 2021 (Wave 1 data); https://doi.org/10.7910/DVN/ZJPKW5, accessed on 4 May 2021 (Wave 2 data).

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
