# Peer review of "Longitudinal Examination of Which Married Young Women Use Contraception to Delay a First Pregnancy in Bihar and Uttar Pradesh, India"

_ijerph, 2023, doi:10.3390/ijerph20156504_

Round 1
Reviewer 1 Report
congratulations for the great efforts spent doing this research. However, the following points need to be addressed.
1) Would you clarify further what is mentioned in line 102-103, i.e. why 5 months were substracted from the day of the survey?
2) What are the traditional methods of contraception
3) table 3 & 4 please correct the word logit to logistic
Author Response
Congratulations for the great efforts spent doing this research. However, the following points need to be addressed.
- Would you clarify further what is mentioned in line 102-103, i.e. why 5 months were substracted from the day of the survey?
Response: Thank you, we have clarified that this was to get an estimate of the timing of the pregnancy, to compare to timing of first contraceptive use. Since the questionnaire did not have the number of months pregnant for the currently pregnant women, we estimated that on average currently pregnant women were about 5 months pregnant.
- What are the traditional methods of contraception
Response: This has been added to the text.
- table 3 & 4 please correct the word logit to logistic
Response: This change has been made.
Reviewer 2 Report
Dear Editor,
Thank you for the opportunity to review the manuscript titled “Which married adolescents use contraception to delay a first pregnancy in Bihar and Uttar Pradesh, India? This paper makes an important contribution to the literature on contraceptive use among adolescents. The setting is also important as if focuses on states that have lagged behind on a number of important indicators, thus underscoring the importance of social development to adolescent health. The paper is well written, utilizes a clear and sound methodology. I take note that the authors verified the results of their hazard analysis with logistic analyses, given that the prior analyses had not used sampling weights. The conclusions are also consistent with their analysis and the literature on contraceptive use in low and middle-income countries.
My comment is minor and has to do with the role of empowerment and intimate partner violence, which are relevant in this highly patriarchal setting. The authors only included one measure of empowerment – educational decision-making. Many prior studies, drawing on household surveys, have included more measures of empowerment such as health and financial decision-making. Also, no measures of violence were included yet the literature suggests that young women who are married earlier, are more at risk for intimate partner violence, which could affect their contraceptive decision-making. It would be helpful if the authors provided some insight on why these variables were included from the analyses. Otherwise, this is an excellent paper.
Author Response
Thank you for the opportunity to review the manuscript titled “Which married adolescents use contraception to delay a first pregnancy in Bihar and Uttar Pradesh, India? This paper makes an important contribution to the literature on contraceptive use among adolescents. The setting is also important as if focuses on states that have lagged behind on a number of important indicators, thus underscoring the importance of social development to adolescent health. The paper is well written, utilizes a clear and sound methodology. I take note that the authors verified the results of their hazard analysis with logistic analyses, given that the prior analyses had not used sampling weights. The conclusions are also consistent with their analysis and the literature on contraceptive use in low and middle-income countries.
My comment is minor and has to do with the role of empowerment and intimate partner violence, which are relevant in this highly patriarchal setting. The authors only included one measure of empowerment – educational decision-making. Many prior studies, drawing on household surveys, have included more measures of empowerment such as health and financial decision-making. Also, no measures of violence were included yet the literature suggests that young women who are married earlier, are more at risk for intimate partner violence, which could affect their contraceptive decision-making. It would be helpful if the authors provided some insight on why these variables were included from the analyses. Otherwise, this is an excellent paper.
Response: Thank you for this excellent comment. In earlier versions of the models, we included additional agency. empowerment and violence variables; however, none of them were significant and thus it was determined that for model simplicity, these variables would be removed. In addition, these variables were not time dependent and thus it was unclear if reports on violence and/or empowerment for Cohort A were relevant to the time of marriage and prior to the first pregnancy/first contraceptive use. The educational decision-making variable seemed to be more related to the time before the young person would have married. We have included this in the limitations and point to the need for future longitudinal studies to assess the role of these types of variables.
Reviewer 3 Report
Which married adolescents use contraception to delay a first pregnancy in Bihar and Uttar Pradesh, India?
Speizer etal
Thank you for considering me as a potential reviewer for this paper. This paper aims to evaluate the factors contributing to contraceptive use among young, married women in Bihar and Uttar Pradesh, India, before their first pregnancy. This is an important topic that could help save the lives of young married women in India. The paper is well-written, and the mathematical models are well-explained, but there is room for improvement to make it more understandable for Public Health audiences. Here are my suggestions.
1. Study title:
Although reflecting the research objectives, it gives the impression of a qualitative study. I suggest adding the study design to be informative.
2. Abstract:
Needs to provide more information about the study, especially the methods. For example, what is this longitudinal data, and time frame? Etc. Regarding the results, it mentioned that the use of contraception before a first pregnancy was low":; what percentage?.
3. Introduction:
- Please be specific when mentioning facts for example, “a small percentage were using contraception” please mention the percentage. (Line 56)
- Please write the name of the survey “Data collected in 2015-2016 among” (line 59).
- Although there is recent literature on the research topic, the authors utilised the reference (Jejeebhoy, et al 2014)* three times in the introduction. It would be helpful to update the introduction with more current studies.
* One of the research team: Jejeebhoy SJ, Santhya KG, Zavier AF. Demand for contraception to delay first pregnancy among young married women in India. Studies in family planning. 2014 Jun;45(2):183-201.
4. Methods:
The current methodology section is difficult to follow. The section on study variables needs to be clarified and trimmed. Finally, in the statistical analysis section, please specify the statistical software utilized for data analysis, and indicate the cutoff level of significance.
5. Results:
- Tables 1 and 2 all comparisons showed significant differences; what about the effect size? Authors should calculate and report effect size measures. P-values are highly correlated with the size of the sample (see Lakens, 2022: Sample size justification). The authors present results on a fairly large sample size. Therefore, most of the tested differences would be significant by default. Therefore, I highly recommend that the authors report effect size measures (which are less biased than p-values).
- In Table 2 is written “All means and percentages”. No means is shown in the Table; please review.
- Table 3 and 4 Please provide the Odds Ratio (OR) along with the regression coefficients it is easier to interpret for Journal reader for example “women from Bihar were less likely to use before a first pregnancy than women from Uttar Pradesh [ OR= 0.47, *p≤0.001].
-
- 6. Discussion:
- Regarding the discussion section, I recommend updating it. It mainly includes reporting the results section while it should comparing with previous studies; only two references have been consulted, one of them is previously mentioned (Jejeebhoy, et al 2014)*
- Study limitation, although the authors mentioned important limitations, variables included in the analysis limited to survey available data, other important confounder were not included in this analysis.
7. Conclusion:
Well prepared
Author Response
Thank you for considering me as a potential reviewer for this paper. This paper aims to evaluate the factors contributing to contraceptive use among young, married women in Bihar and Uttar Pradesh, India, before their first pregnancy. This is an important topic that could help save the lives of young married women in India. The paper is well-written, and the mathematical models are well-explained, but there is room for improvement to make it more understandable for Public Health audiences. Here are my suggestions.
- Study title:
Although reflecting the research objectives, it gives the impression of a qualitative study. I suggest adding the study design to be informative.
Response: The title has been changed.
- Abstract:
Needs to provide more information about the study, especially the methods. For example, what is this longitudinal data, and time frame? Etc. Regarding the results, it mentioned that the use of contraception before a first pregnancy was low":; what percentage?.
Response: These changes have been made.
- Introduction:
- Please be specific when mentioning facts for example, “a small percentage were using contraception” please mention the percentage. (Line 56)
Response: This has now been added to the text.
- Please write the name of the survey “Data collected in 2015-2016 among” (line 59).
Response: This has been added.
- Although there is recent literature on the research topic, the authors utilised the reference (Jejeebhoy, et al 2014)* three times in the introduction. It would be helpful to update the introduction with more current studies.
* One of the research team: Jejeebhoy SJ, Santhya KG, Zavier AF. Demand for contraception to delay first pregnancy among young married women in India. Studies in family planning. 2014 Jun;45(2):183-201.
Response: Thank you for this observation. We have now added a few additional references to the introduction (and discussion) to make these sections more current. These references are highlighted in yellow on the reference list.
- Methods:
The current methodology section is difficult to follow. The section on study variables needs to be clarified and trimmed. Finally, in the statistical analysis section, please specify the statistical software utilized for data analysis, and indicate the cutoff level of significance.
Response: The variables section has been simplified based on this reviewer and the other reviewers’ comments. We have added the statistical software used for data analysis. We have also mentioned the significance levels presented.
- Results:
- Tables 1 and 2 all comparisons showed significant differences; what about the effect size? Authors should calculate and report effect size measures. P-values are highly correlated with the size of the sample (see Lakens, 2022: Sample size justification). The authors present results on a fairly large sample size. Therefore, most of the tested differences would be significant by default. Therefore, I highly recommend that the authors report effect size measures (which are less biased than p-values).
Response: In Table 1, only a small number of the comparisons between Cohort A and Cohort B are significantly different. This is not unexpected as the sample comes from the same sites. For Table 2, there are more differences which are not surprising given that Cohort A was married longer (and at a younger age). We feel that the way the results are presented in Tables 1 and 2 is clear and transparent of the extent of differences between the groups as the overall percentages for each group can be compared and it is possible to see that when there are significant differences, the size of the difference is of reasonable magnitude (e.g., at least 5-10% difference between groups).
- In Table 2 is written “All means and percentages”. No means is shown in the Table; please review.
Response: Good catch, thank you this has been dropped.
- Table 3 and 4 Please provide the Odds Ratio (OR) along with the regression coefficients it is easier to interpret for Journal reader for example “women from Bihar were less likely to use before a first pregnancy than women from Uttar Pradesh [ OR= 0.47, *p≤0.001].
Response: Thank you for pointing this out. We have changed the Table 3 and 4 results to be odds ratios and 95% confidence intervals and Table 5 and 6 to hazard ratios and significance levels. These are not shown with track changes but rather highlighted since all of the values changed. We have also changed how we talk about this in the text.
- 6. Discussion:
- Regarding the discussion section, I recommend updating it. It mainly includes reporting the results section while it should comparing with previous studies; only two references have been consulted, one of them is previously mentioned (Jejeebhoy, et al 2014)*
Response: Thank you for this excellent comment. We have revised the discussion section (and introduction) and now include reference/discussion of additional relevant studies.
- Study limitation, although the authors mentioned important limitations, variables included in the analysis limited to survey available data, other important confounder were not included in this analysis.
Response: This is an excellent point and related to the comment from Reviewer 2. We have added an additional limitation to this section related to these points.
- Conclusion:
Well prepared
Response: Thank you.
Round 2
Reviewer 3 Report
Thank you, and good luck with your excellent paper!
Author Response
Thank you for the positive comments on the paper. No specific revisions were requested here.